# Influence of Natural Variability and Anatomical Misalignment on the Correlation Between Segmental Myocardial Edema and Strain in Acute Myocarditis

**DOI:** 10.3390/biomedicines13030712

**Published:** 2025-03-14

**Authors:** Kanza Awais, Lana Kralj, Andreja Cerne Cercek, Borut Kirn

**Affiliations:** 1Institute of Physiology, Faculty of Medicine, University of Ljubljana, 1000 Ljubljana, Slovenia; 2Department of Cardiology, University Medical Centre Ljubljana, 1000 Ljubljana, Slovenia

**Keywords:** speckle tracking echocardiography, segmental peak systolic strain, cardiac magnetic resonance, late gadolinium enhancement, acute myocarditis, natural heterogeneity, anatomical orientation mismatch, spatial resolution reduction

## Abstract

**Background:** Acute myocarditis (AM) affects myocardial structure and function, assessed by cardiac magnetic resonance late gadolinium enhancement (CMR-LGE) and speckle tracking echocardiography (STE), respectively; however, the correlation between the two techniques at the segmental level is inconsistent. We studied natural heterogeneity and anatomical orientation mismatch as potential causes of correlation discrepancy. **Methods:** A total of 30 AM patients underwent left ventricle LGE-CMR and STE measurement, acquiring 18 segmental values depicting edema extent and peak longitudinal strain, respectively. Baseline segmental correlation was compared to average patient segmental correlation and to segmental correlation after spatial resolution reduction achieved by averaging adjacent segments in four successive iterations, where the degree of spatial resolution reduction was evaluated based on the relative decrease in segmental standard deviation. **Results:** Baseline segmental correlation was weak, i.e., r = 0.24 (*p* < 0.05) but improved in fitted SLGE and SpLS baseline correlation (r_0_ = 0.44, *p* < 0.05) and in average patient correlation (r = 0.55, *p* < 0.05). Iterative spatial resolution reduction increased the correlation to r_1_ = 0.49 and r_2_ = 0.51 and then decreased it to r_3_ = 0.11 (*p* < 0.05) and r_4_ = 0.07 (*p* > 0.05), with corresponding decreases in segmental standard deviation relative to baseline from σ_0_ = 12.87 to σ/σ_0_ = 0.68, 0.51, 0.38, and 0.29 in SLGE values and σ_0_ = 4.77 to σ/σ_0_ = 0.57, 0.41, 0.31, and 0.23 in SpLS. **Conclusions:** Improved correlation in average patients is associated with natural heterogeneity, which indicates a need to develop more robust indicators of ventricular function. The improved correlation in moderate spatial resolution reduction indicates a potential solution for anatomic orientation mismatch between CMR-LGE and STE techniques.

## 1. Introduction

Acute myocarditis (AM) is the inflammation of the myocardium with a wide range of clinical manifestations, often inducing complex alterations in the ventricular structure, leading to dysfunction [1]. Diagnosis of the disease seldom relies only on biochemical markers and conventional techniques. To confirm the diagnosis, cardiac magnetic resonance (CMR) has become the gold standard among noninvasive imaging techniques for cardiac tissue characterization specifically via early and late gadolinium enhancement (LGE) [2]. It enables segmental quantification of myocardial inflammation, thus providing its location and extent. Due to high cost and time-consuming nature of CMR-LGE, it is rarely performed in follow-ups. Echocardiography has become a useful initial diagnostic and follow-up tool for patients with suspected AM [3,4]. It is typically performed to visualize functional abnormalities. Speckle tracking echocardiography (STE), based on postprocessing of echocardiographic cine recordings, is becoming a widely available technique that enables quantitative assessment of segmental myocardial mechanics, including strain and strain rate in longitudinal, radial, and circumferential dimensions. Compared to echocardiography, STE can quantify regional changes, offering a more detailed evaluation of both the location and extent of myocardial dysfunction.

While CMR-LGE effectively detects myocardial structural abnormalities and provides tissue characterization [3], STE has been found to be a diagnostic marker for assessing functional impairment in AM. Uppu et al. demonstrated reductions in both global and segmental peak systolic strain in patients of AM [5], findings that have been validated by other studies [6,7,8].

However, when correlating the structural and functional changes between CMR-LGE and STE, conflicting findings have been observed across global and segmental levels. While global analysis has shown good correlation between global longitudinal strain (GpLS) and edema extent [9,10,11], findings at the segmental level are more variable. A moderate correlation has been found at the segmental level in a study by Uppu et al. [5], whereas Kandels et al. observed that strain impairment does not consistently align with LGE presence in the same segments but in the neighboring ones [6]. Conversely, some researchers have found no correlation [12], leaving the segmental correlation data relatively sparse and inconsistent.

The segmental LGE and STE measurements show, at best, only moderate correlation, indicating that segmental LGE measurements cannot be replaced by segmental STE measurements, which would be a preferrable technique due to its low cost and ambulatory availability. This study aims to explore the potential causes of the weak correlation between the two techniques, focusing on natural heterogeneity and anatomical orientation misalignment.

## 2. Material and Methods

### 2.1. Patients

From January 2017 to October 2020, we retrospectively analyzed the data of thirty cases of AM patients who underwent CMR-LGE and echocardiography. The suspicion on AM was confirmed by the following clinical symptoms indicative of AM: increased troponin I levels, “myocarditis-like” MRI findings, electrocardiography (ECG) changes suggestive of AM, and absence of coronary artery disease by coronary angiography. This study was conducted in accordance with the ethics committee requirements for retrospective analysis of anonymized clinical images and approved by the local ethics committee (resolution no. 0120-37/2017/4 on 27 November 2017).

### 2.2. Echocardiography

Echocardiography was performed on each patient by an experienced cardiologist using a GE vivid E95 echocardiographic machine (Horten, Norway) equipped with a 4Vc phased-array probe to acquire functional left ventricular (LV) parameters. To cover the entire LV in longitudinal dimension, cine recordings were acquired in the apical 4, 3, and 2 chamber views. STE was performed on the cine recordings of these patients by the same cardiologist using EchoPAC software (version 206, GE Healthcare, Horten, Norway) and the Q-analysis application. The myocardial border was manually traced, and strain patterns were acquired with the reference value set at end-diastole, which was determined from aortic valve opening (AVO) and the ECG—R wave. SpLS values were obtained as the maximum value of the peak negative strain during the systole from all the planes in 18 sectors (6 in the apex, mid, and base regions) distributed across the entire LV wall. These SpLS values were averaged to acquire GpLS. Figure 1 shows an example of STE-derived segmental strain patterns, with each pattern highlighted in different colors, representing one specific segment of the LV and peak strains (pLS) (Figure 1a) along with a bull’s eye plot of SpLS from case no. 10 (Figure 1b).

### 2.3. Cardiovascular Magnetic Resonance Imaging (CMR)

CMR examination was conducted on all the patients using a 3 Tesla (T) Siemens TrioTim scanner (Siemens Healthineers, Forcheim, Germany). ECG-gated steady-state free precession cine images were acquired in the standard short-axis slices covering the entire left ventricle. Before the imaging process, patients received an intravenous injection of 0.1 mmol/kg of Gadovist, a contrast agent containing gadolinium. LGE imaging was then achieved by phase-sensitive inversion recovery (PSIR) imaging sequences covering the entire LV in short axis. The images were T1-weighted, with standard settings including a repetition time of 700–800 ms, echo time of 1.5–1.6 ms, and delay time of 500–650 ms. The matrix size was 256 × 256, the flip angle was set at 20°, and the in-plane resolution was 1.4 × 1.4 mm^2^. The inversion time was adjusted to eliminate the signal from normal myocardium, typically set between 250 and 350 ms. Images were taken from 10 to 12 short-axis planes to ensure full coverage of LV from the base to the apex, with each slice having a thickness of 8 mm and inter-slice spacing of 8 to 10 mm.

### 2.4. LGE Quantification

Image analysis and LGE quantification were performed using specialized software [13]. Initially, the cardiologists manually delineated the epicardial and endocardial border of the LV during the end-diastolic phase. The extent of LGE was then quantified using the 3σ mathematical quantification method, which applies a signal intensity threshold set at three standard deviations above the mean signal intensity of normal myocardium to identify pathological regions. Manual corrections were applied to exclude pathology-free regions and ensure that visible artifacts appearing bright on the images were not included in the calculations (Figure 2a).

SLGE values were obtained using 16-segment bull’s eye plots, which visually represent the distribution of pathology across the entire heart level in a single graph, as illustrated in Figure 2b. Each plot is composed of apical, mid, and basal regions depicting LV segments and is oriented with respect to the anterior point.

Prior to analysis, both pLS and LGE data were preprocessed to remove outliers. Segmental values deviating beyond three standard deviations (SDs) from the mean were considered non-physiological. However, since all segmental values fell within this range, none of the segments were excluded from the analysis.

The four segments in the apical region of the CMR-LGE bull’s eye plots were then linearly interpolated into six segments. The orientation of the CMR-LGE data was manually adjusted to align with the STE bull’s eye plot, using the reference point marked by an arrow in Figure 1 and Figure 2. This reference point corresponds to the midpoint of the anterior septal segment on the 18-segment bull’s eye plot.

### 2.5. Spatial Resolution Reduction Approach

A process of spatial resolution reduction included equal weighted averaging of adjacent segments on the bull’s eye plot. In the basal and apical regions, the observed segment and its three adjacent segments were averaged, while in the mid region, the observed segment and its four adjacent segments were averaged. This process was repeated over four iterations for both SLGE and SpLS values. At each iteration, we calculated the segment-to-segment correlation coefficient (r) between SLGE and SpLS along with the standard deviation (σ) relative to the baseline standard deviation (σ_0_) of all segmental values for each patient. We observed a consistent decrease in σ with respect to the baseline σ_0_ with each successive iteration, indicating a progressive reduction in spatial resolution compared to its initial value.

### 2.6. Correlation

As a baseline, SLGE and SpLS values were correlated across all segments for each patient, and the resulting correlation coefficients were averaged. This baseline value was then compared with correlations obtained using two distinct approaches. In the first approach, we averaged the SLGE and SpLS values across all patients for each segment and then calculated the correlation between the averaged SLGE and SpLS segments. In the second approach, we correlated SLGE and SpLS values within the same segments for each patient, after four iterative reductions in spatial resolution and the resulting correlation coefficients were averaged across the patient group.

### 2.7. Statistical Analysis

Statistical analysis was conducted using MATLAB R2024a and Microsoft Excel. To account for the hierarchical structure of myocardial segments nested within patients and lack of statistical independence, a linear mixed-effects model was applied instead of the traditional nested ANOVA. In this model, SLGE and SpLS were treated as fixed effects, while patients were modeled as random intercepts to account for intra-patient variability. The model was implemented using the fitlme function in MATLAB with restricted maximum likelihood estimation to obtain unbiased variance estimates. The model structure was as follows: SLGE_ij_ = β_0_ + β_1_. SpLS_ij_ + u_i_ + ϵij, where SLGE_ij_ represents the SLGE for segment j in patient I, SpLS_ij_ represents the fixed-effect predictor for the corresponding segment, β_0_ represents the intercept, β_1_ represents the fixed-effect coefficient, u_i_ represents the patient-specific random effect, and ϵij represents the residual error [14,15].

Normality assumptions were tested using the Kolmogorov–Smirnov (K-S) test at the baseline, at the average patient level, and in each successive iteration for SLGE and SpLS. For the K-S test, a *p* value ≥ 0.05 was considered indicative of normality. Data for the continuous variables were presented as means ± standard deviation, while the categorical variables were presented as percentages.

For comparison involving normally distributed data, Student’s t-test was used. Correlation analysis was performed based on the normality of data: Pearson correlation was used for variables which met the assumptions of normality, while Spearman’s rank correlation was employed in cases where normality assumptions were violated. The correlation coefficient (r) was reported for both Pearson and Spearman’s rank correlations, with statistical significance defined as *p* ≤ 0.05. Additionally, the sample size was selected based on previous studies [4,9,16]

## 3. Results

The patient cohort included thirty patients with AM. Both echocardiography and CMR exam were performed either at the time of diagnosis or one day later. Baseline characteristics of all the patients are represented in Table 1. Patients had a median troponin I level of 12,755 ng/L (range: 129–146,000 ng/L; normal < 34 ng/L) and a median C-reactive protein level of 35.5 mg/L (range: <5–260 mg/L). Echocardiography showed an LV ejection fraction greater than 52%. However, all patients presented myocardial edema with an extent of 17.7 ± 17.3% on LGE-CMR.

### 3.1. Baseline Correlation and Hierarchial Variance Analysis

The K-S test showed that the data were non-normally distributed, so Spearman’s rank correlation was applied to assess the relationship between SLGE and SpLS. The result showed a weak, statistically significant relation with r = 0.24 (*p* < 0.05). The nested linear mixed-effects model applied to remove pseudoreplication revealed a significant relationship between SLGE and SpLS. For the SLGE model, SpLS significantly predicted SLGE, with an F = 17.36 (*p* < 0.05). For the SpLS model, SLGE was a significant predictor of SpLS, with an F = 16.53 (*p* < 0.05). Both models showed substantial variability in random effects at both patient and segmental levels.

The correlation between the fitted values of SLGE and SpLS from linear mixed-effects analysis was r_0_ = 0.44 (*p* < 0.05). A scatter plot displaying all 30 × 18 pairs of fitted SLGE and SpLS values is shown in Figure 3.

### 3.2. Cohort-Averaged Segmental Correlation

When the patient cohort was averaged across each segment, the correlation improved significantly, demonstrating a moderate, statistically significant relationship with r = 0.55 and *p* < 0.05. Figure 4 illustrates this correlation, where each point on the graph represents the cohort-averaged SLGE and corresponding SpLS value for all the myocardial segments.

### 3.3. Spatial Resolution Reduction Correlation

While the correlation coefficient (r) progressively improved till the second iteration of spatial resolution reduction, the σ of SLGE and SpLS continuously decreased for each subject. Starting from a baseline correlation of r_0_ = 0.44 (*p* < 0.05) between fitted SLGE and SpLS, the correlation improved to a moderate and statistically significant level of r_1_ = 0.49 (*p* < 0.05) in the first iteration and r_2_ = 0.51 (*p* < 0.05) in the second iteration. In the third and fourth iterations, this correlation started decreasing to r_3_ = 0.11 (*p* < 0.05) and r_4_ = 0.07 (*p* > 0.05), respectively.

The relative decrease in σ ranged from baseline σ_0_ = 12.87 to σ/σ_0_ = 0.68, 0.51, 0.38 to 0.29 across successive iterations for SLGE, whereas for SpLS, σ_0_ = 4.77 decreased to σ/σ _0_ = 0.57, 0.41, 0.31, and 0.23 by the fourth iteration.

Figure 5 illustrates the consequences of spatial resolution reduction iterations on the distribution of SLGE and SpLS in bull’s eye plots, using data from case no. 26. Notably, the correlation coefficients progressively increased, and σ decreased gradually and consistently across each iteration.

It seems that with each successive iteration, the correlation increases exponentially, while the spatial resolution decreases linearly.

## 4. Discussion

We found a weak correlation between SLGE and SpLS raw values. However, both SLGE and SpLS show a strong influence on each other based on the adjusted random-effects model. The correlation between the fitted SLGE and SpLS can be increased by two approaches: (1) averaging the patient cohort for each segment and (2) successive iterations of spatial resolution reduction.

Our segmental correlation results align with previous studies, where qualitative segmental analysis also yielded a weaker correlation [5,6,9]. Previous studies, such as those by Erley and Logstrup et al., have suggested that weak segmental correlations may stem from inconsistencies across individual patients, as well as intrinsic physiological differences [9,17]. Thus, the reason might be natural heterogeneity in the population, as inter-patient variability could introduce noise and reduce the signal-to-noise ratio [18]. The improvement in correlation observed after averaging the patient cohort across segments confirms that the likely cause is indeed randomly distributed natural variability.

Moreover, CMR and echocardiography are two different imaging modalities, with their intrinsic methodological and modality specific discrepancies further contributing to the difficulty in precisely matching functional strain data with the distribution of LGE [19]. Reducing spatial resolution through iterative averaging resulted in an increased correlation from 0.44 at baseline to 0.51 in the second iteration, suggesting the presence of intrinsic misalignment, which persists despite careful alignment of anatomical features between the two measurement modalities. This issue arises from differences in anatomical orientation and spatial resolution between both modalities, leading to mismatched alignments between STE and LGE segments, which has also been reported by Mirea et al. [20]. Leitman et al. emphasized segmental misalignment as a primary obstacle to accurate STE-LGE comparison [21], and similar observations from other researchers highlight that poor spatial correlation can compromise the reliability of aligning strain and LGE data [9,17]. While our approach contributed to reducing this discrepancy, improvement in correlation comes with a trade-off, as lowering spatial resolution reduces regional specificity, which is important for understanding localized myocardial pathology. The observed reduction in σ—from a raw value of 12.87 at baseline to a relative value of 0.29 by the fourth iteration for SLGE and from a raw value of 4.77 to the relative value of 0.23 by the fourth iteration for SLS—confirms that this approach produces more uniform data, even though at the potential expense of reduced diagnostic specificity. We found little benefit when iterating beyond the second iteration, as the correlation coefficient and σ started reducing.

## 5. Study Limitations

A limitation is that it is a retrospective study with a small number of subjects due to data collection from a single center. Echocardiographic recordings were processed with EchoPAC Q analysis rather than AFI, and manual delineation of the myocardial border was performed. In addition, manual delineation of the LV borders was also performed on LGE-CMR images. Manual delineation is intrinsically prone to variability, posing a challenge in both strain and LGE quantification. In our previously published study, we analyzed interobserver and intraobserver variabilities in LGE quantification using the 3σ method, finding interobserver intraclass correlation coefficients (ICCs) of 0.56 (95% CI: 0.15–0.86) and intraobserver ICCs ranging from 0.22 to 0.60 [13]. Similarly, in peak longitudinal strain measurements, intra- and interobserver variabilities ranging from 0.9 to 1.7% were reported across nine echocardiographic devices [22].

Furthermore, the use of different imaging planes (short-axis LGE-CMR and long-axis STE) and segmental models (16 segment vs. 18 segment) could have introduced heterogeneity in the results. Despite these differences, our findings of weak correlation are consistent with previous studies [6,12].

Another potential drawback in comparing SLGE and SpLS is that while SLGE represents a cumulative measure of segmental tissue inflammation, pLS as a single point on the strain curve may lack the sensitivity to fully capture localized myocardial changes reflected in LGE findings. Peak strain, as a single time-point measure, might overlook dynamic aspects of myocardial dysfunction that are critical for correlating functional impairment with tissue characteristics. Future studies should investigate alternative STE-derived parameters, which might correlate more strongly with LGE and provide a more subtle understanding of myocardial tissue changes in AM.

## 6. Conclusions

This study revealed that the weak correlation between segmental LGE and STE is influenced by at least two factors: natural heterogeneity and mismatch in anatomical orientation. While there is very little one could do to reduce natural heterogeneity when deriving segmental LGE-CMR values from measured segmental STE values apart from developing less sensitive indicators in the future, the influence of the mismatch in the anatomical orientation can be significantly reduced through two iterations of spatial resolution reduction.

## Figures and Tables

**Figure 1 biomedicines-13-00712-f001:**
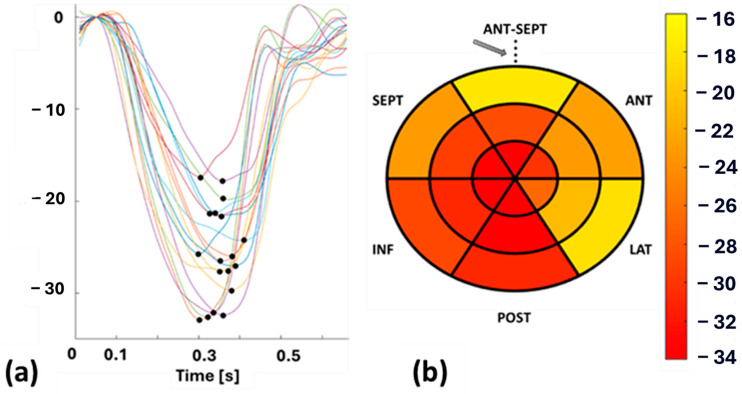
(**a**): Segmental longitudinal strain (SpLS) patterns with peak strain (pLS) indicated by a black dot, and (**b**): bull’s eye plot of SpLS values from case no. 10. Abbreviations: ANT-SEPT—anterior septal, ANT—anterior, LAT—lateral, POST—posterior, INF—inferior, SEPT—septal. The arrow in (**b**) denoted the reference point, which is also the midpoint of the ANT-SEPT region.

**Figure 2 biomedicines-13-00712-f002:**
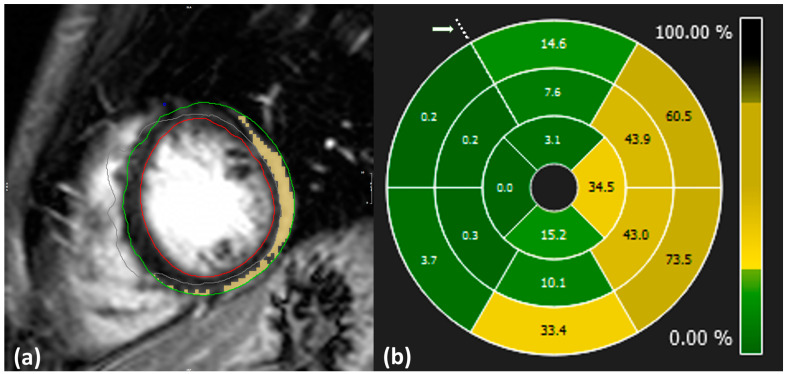
(**a**): Example of LGE quantification from CMR-LGE images. Blue: anterior point of intersection of left ventricle, right ventricle, and septum; green: delineation of epicardium; red: delineation of endocardium; yellow: inflamed region; gray: excluded region. (**b**): Distribution of LGE values across the whole LV level displayed in 16 segment plots from case no. 10. Dotted points indicate the anterior intersection of the right ventricle, left ventricle, and septum as highlighted by an arrow.

**Figure 3 biomedicines-13-00712-f003:**
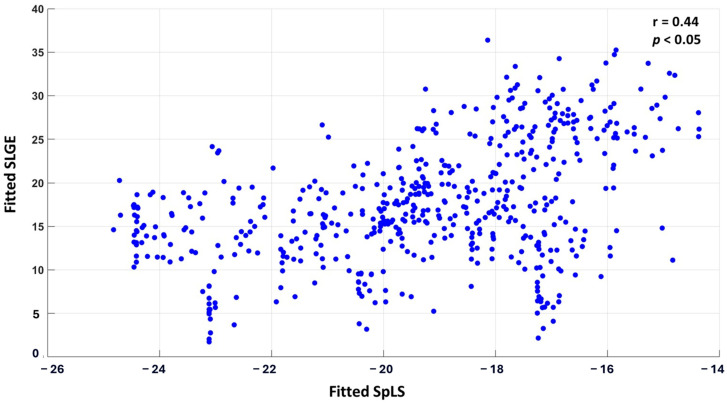
Scatter plot correlating SLGE and SpLS in 540 myocardial segments acquired from 18 segments of 30 patients of AM.

**Figure 4 biomedicines-13-00712-f004:**
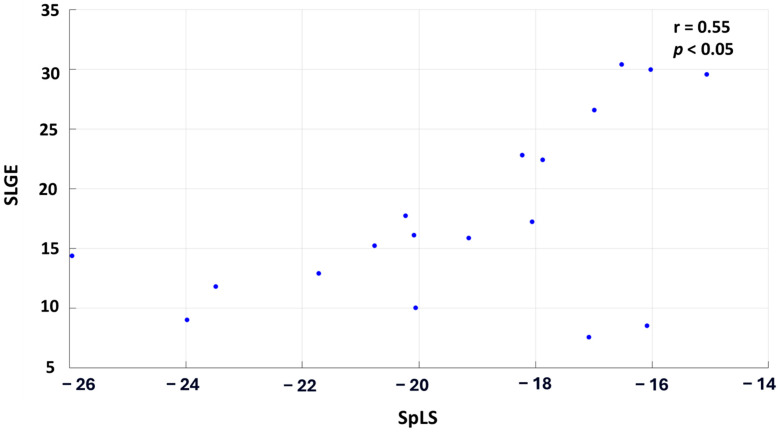
Scatter plot correlating the averaged SLGE and SpLS values across the 18 segments in 30 patients of AM.

**Figure 5 biomedicines-13-00712-f005:**
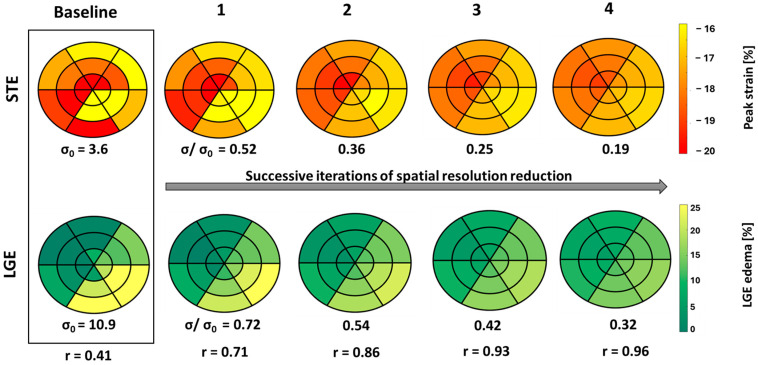
18-segment plots of baseline pLS and LGE values from STE and LGE, respectively, displayed in a box with four successive iterations of spatial resolution reduction, along with their σ and r values from case no. 26. Less negative peak strain (yellow) indicates a higher extent of dysfunction, and higher LGE (yellow) indicates larger edema.

**Table 1 biomedicines-13-00712-t001:** Summary of patient characteristics. Abbreviations: BMI—body mass index, HR—heart rate, CRP—C-reactive protein, STE—speckle tracking echocardiography, EF—ejection fraction, GpLS—global peak longitudinal strain, GLGE—global late gadolinium enhancement.

Baseline Characteristics of Patients	
No. of cases (n)	30
Gender (M/F)	26/4
Age (years)	29.6 ± 7.7
BMI (kg/m^2^)	26.5 ± 5.9
Peak troponin (ng/mL)	19.3 ± 28.4
CRP elevated (ng/L)	58.4 ± 63
STE EF (%)	62 ± 8.9
STE GpLS (%)	−19.3 ± 5.5
GLGE extent (%)	17.7 ± 17.3

## Data Availability

All data supporting the findings of this study are included in the article; any additional inquiries can be directed to the corresponding author.

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
