# Peer review of "Influence of Natural Variability and Anatomical Misalignment on the Correlation Between Segmental Myocardial Edema and Strain in Acute Myocarditis"

_biomedicines, 2025, doi:10.3390/biomedicines13030712_

Round 1
Reviewer 1 Report
Comments and Suggestions for Authors
This paper has clear ideas and is meaningful to explore the factors of weak correlation between myocardial LGE and STE. But there are still some issues that need to be clarified.
Research sample size: 30 subjects were used. Is this sample size sufficient? The calculation process of the sample volume should be explained in the statistics section.
Statistical method: Through the display of the mean addition and subtraction of the standard deviation, is the data normally distributed for small samples?
In which time phase is the SLGE 16-segment target map measured? It is crucial whether it can be consistent with the echocardiogram.
Since manual annotation is used in both modes (ultrasound and mri), the variability between readers and intra-readers should be evaluated, which affects the persuasiveness of the results.
Author Response
We thank the reviewers for their positive remarks. The manuscript has been duly reviewed and corrected based on the constructive suggestions by all reviewers, thereby we believe that the final version would represent scientific message more lucidly.
Please find the detailed point by point responses below with all the corrections highlighted in Yellow in the re-submitted files.
Reviewer # 1:
- This paper has clear ideas and is meaningful to explore the factors of weak correlation between myocardial LGE and STE. But there are still some issues that need to be clarified.
Thank you for your appreciation. We will try our best to clarify the confusions addressed.
- Research sample size: 30 subjects were used. Is this sample size sufficient? The calculation process of the sample volume should be explained in the statistics section.
We acknowledge your concern regarding the sample size of 30 subjects. While we agree that this sample size is relatively small, it reflects the number of subjects meeting the inclusion criteria of our study from a single center during data collection period. Although a larger sample size would have allowed for more robust analysis, our sample size is comparable to previously conducted similar studies with references mentioned in the statistics section (Line no. 167) and below.
- Goody, P. R., Zimmer, S., Öztürk, C., Zimmer, A., Kreuz, J., Becher, M. U., Isaak, A., Luetkens, J., Sugiura, A., Jansen, F., Nickenig, G., Hammerstingl, C., & Tiyerili, V. (2022). 3D-speckle-tracking echocardiography correlates with cardiovascular magnetic resonance imaging diagnosis of acute myocarditis - An observational study. International journal of cardiology. Heart & vasculature, 41, 101081. https://doi.org/10.1016/j.ijcha.2022.101081
- Løgstrup, B. B., Nielsen, J. M., Kim, W. Y., & Poulsen, S. H. (2016). Myocardial oedema in acute myocarditis detected by echocardiographic 2D myocardial deformation analysis. European heart journal. Cardiovascular Imaging, 17(9), 1018–1026. https://doi.org/10.1093/ehjci/jev302
- Meindl, C., Paulus, M., Poschenrieder, F., Zeman, F., Maier, L. S., & Debl, K. (2021). Patients with acute myocarditis and preserved systolic left ventricular function: comparison of global and regional longitudinal strain imaging by echocardiography with quantification of late gadolinium enhancement by CMR. Clinical research in cardiology : official journal of the German Cardiac Society, 110(11), 1792–1800. https://doi.org/10.1007/s00392-021-01885-0
- Statistical method: Through the display of the mean addition and subtraction of the standard deviation, is the data normally distributed for small samples?
To assess the normality of our SLGE and SpLS values, we performed Kolmogorov-Smirnov test, and the data was normally distributed. This has been added to the Statistical analysis section, line no. 162.
- In which time phase is the SLGE 16-segment target map measured? It is crucial whether it can be consistent with the echocardiogram.
SLGE measurements were performed during the end – diastolic phase of cardiac cycle as this phase provides most stable representation of myocardial segmental anatomy based on standard practice in cardiac imaging (Methods section, Paragraph 4, Line 115).
ECG gating was used during both CMR and echocardiography to ensure consistency of same physiological timepoints.
- Since manual annotation is used in both modes (ultrasound and mri), the variability between readers and intra-readers should be evaluated, which affects the persuasiveness of the results.
We acknowledge that the variability between readers and within the same reader can affect the reliability of the results. To address this, we have included a paragraph in the limitation section discussing the influence of intra reader and inter reader variability on the measurements (Line 261 – 267)
Reviewer 2 Report
Comments and Suggestions for Authors
This is an interesting article about imaging in myocarditis. The article may be published, as long as the authors make some needed changes, including:
- what statistical software was used
- results - not all patients had a median troponin, but rather the patients
- table 1 - is the average BMI correct? 47 is quite high for an average
- figure 3 - what do the p and r represent? which statistical test? similar for other figures
- the number of references is very low; discussion should be enriched with more comparisons with similar studies.
Author Response
We thank the reviewers for their positive remarks. The manuscript has been duly reviewed and corrected based on the constructive suggestions by all reviewers, thereby we believe that the final version would represent scientific message more lucidly.
Please find the detailed point by point responses below with all the corrections highlighted in Yellow in the re-submitted files.
Reviewer # 2:
- This is an interesting article about imaging in myocarditis. The article may be published, as long as the authors make some needed changes, including:
Thank you for your positive remarks. We will try our best to clarify all the points addressed.
- what statistical software was used
The statistical softwares used in this study are MATLAB R2024a and Microsoft excel which have been mentioned in the statistical analysis section (Line 161).
- results - not all patients had a median troponin, but rather the patients
We agree. Thank you for highlighting. We have repaired this sentence as highlighted in yellow (Line 173)
- Table 1 - is the average BMI correct? 47 is quite high for an average
I apologize for this typo error that we have corrected now. The average BMI is 26.5 ± 5.9 kg/m2.
- figure 3 - what do the p and r represent? Which statistical test? similar for other figures
The statistical test employed in this study is Person correlation analysis. This has been added with the clear description of r and p values in the result section (Line 183, 184, 190, 191, 201 – 206). For more clarity, we have defined the statistical test with r and p values in the statistical analysis section as well (Line 164, 165)
- The number of references is very low; discussion should be enriched with more comparisons with similar studies.
The no. of studies directly addressing the specific issue of weak segmental correlation between CMR-LGE and STE is low. In the context of natural heterogeneity and anatomical orientation misalignment, relevant references are nearly non-existent. However, we have now expanded the discussion by incorporating a few additional references reflecting similar challenges in correlating strain and LGE (Line 227, 234 -236, 242).
Reviewer 3 Report
Comments and Suggestions for Authors
Dear authors,
Your study is very interesting and I am sure it will contribute to the field of biomedicine. However, I will make some comments and adjustments before its publication.
1. In the statistical analysis section you indicate that the technique you applied was Pearson correlation and a substantive concern is that you do not indicate whether you verified the assumptions inherent to the test, for example, normal error distribution of the variables. Please verify and indicate the test with which you accept or reject the set of hypotheses associated with the test, in addition to showing the values ​​in the statistical analysis section.
2. According to the representation in Figure 1, 16 segmental longitudinal strains are shown and these are grouped into six sections. In this sense, and I consider it to be the most important point, the records represent 16 pseudoreplicas per patient. Therefore, the correlation analysis between SpLs and SLGE shows the associated dispersion but due to the pseudoreplicates a clear pattern associated with the segments is not observed and therefore the Pearson correlation statistical test applied in this way is not functional and neither inferential. There are two ways to analyze this data:
a) Obtain the average values ​​and standard errors for each segment of the 30 patients and graph them in x and y with the associated standard errors for SpLs and SLGE (see paper). Therefore, 16 linear/or non-linear regressions can be estimated to be able to make inferences.
Rensink, R. A. (2017). The nature of correlation perception in scatterplots. Psychonomic Bulletin & Review, 24, 776-797.
b) Perform a nested ANOVA analysis, where the average response can be compared at different levels between the 16 segments and six sections. This analysis resolves the presence of pseudoreplicates, which is a recurring situation for obvious reasons in medical or biomedical studies, although it is not exclusive to the area. To help in this process, the records of the 16 segments must be nested in the six sections and both as simple nesting or double nesting in the patients. Here the reason for the comparative and associated hypothesis games of the ANOVA lies in highlighting the importance and contrast between segments and sections. It is important to indicate that the records according to the statistical model may require that the nesting be random. Annex pdf of article and citation of the book to be reviewed and verify the assumptions of normal error distribution and homogeneity of variances.
Lazic, S. E. (2010). The problem of pseudoreplication in neuroscientific studies: is it affecting your analysis? BMC Neuroscience, 11, 1-17.
Zar, J. H. (2019). Biostatistical analysis. Pearson Education, Incorporated.
Applying this type of analysis is a way of innovating from the methodological perspective, in addition to not incurring errors by applying tests incorrectly and violating the assumptions. In this sense, it also allows the reproducibility of the study and technical repeatability.
3. Check that the references are located in the format requested by the Journal.
Kind regards,
Comments on the Quality of English Language
The English could be improved
Author Response
We thank the reviewers for their positive remarks. The manuscript has been duly reviewed and corrected based on the constructive suggestions by all reviewers, thereby we believe that the final version would represent scientific message more lucidly.
Please find the detailed point by point responses below with all the corrections highlighted in Yellow in the re-submitted files.
Reviewer # 3:
- In the statistical analysis section, you indicate that the technique you applied was Pearson correlation and a substantive concern is that you do not indicate whether you verified the assumptions inherent to the test, for example, normal error distribution of the variables. Please verify and indicate the test with which you accept or reject the set of hypotheses associated with the test, in addition to showing the values ​​in the statistical analysis section.
Thank you for your valuable feedback. We have now included a more detailed explanation of how we verified the assumptions of normality for the Pearson correlation. Specifically, we used Kolmogorov – Smirnov (K – S) test to assess the normal distribution of our continuous variables. The results of the K – S test showed that the majority of the data followed a normal distribution, which supports the use of Pearson correlation for our analysis. The hypotheses associated with the Pearson correlation were tested based on assumptions of normality, and statistical significance was defined as a p - value ≤ 0.05.
Furthermore, the statistical approach used in this study is consistent with the methodologies employed in the following studies, where correlation was performed in a similar manner:
- Spartera, M., Damascelli, A., Mozes, F., De Cobelli, F., & La Canna, G. (2017). Three-dimensional speckle tracking longitudinal strain is related to myocardial fibrosis determined by late-gadolinium enhancement. The international journal of cardiovascular imaging, 33(9), 1351–1360. https://doi.org/10.1007/s10554-017-1115-1
- Leitman, M., Vered, Z., Tyomkin, V., Macogon, B., Moravsky, G., Peleg, E., & Copel, L. (2018). Speckle tracking imaging in inflammatory heart diseases. The international journal of cardiovascular imaging, 34(5), 787–792. https://doi.org/10.1007/s10554-017-1284-y
- Ye, J., Zong, W., Wu, X., Shao, X., & Wu, Y. (2023). Quantitative evaluation of acute myocardial infarction by feature-tracking cardiac magnetic resonance imaging. Pakistan journal of medical sciences, 39(3), 804–808. https://doi.org/10.12669/pjms.39.3.7248
- Uppu SC, Shah A, Weigand J, Nielsen JC, Ko HH, Parness IA, Srivastava S. Two-dimensional speckle-tracking-derived segmental peak systolic longitudinal strain identifies regional myocardial involvement in patients with myocarditis and normal global left ventricular systolic function. Pediatr Cardiol. 2015 Jun;36(5):950-9. doi: 10.1007/s00246-015-1105-9. Epub 2015 Jan 24. PMID: 25617227.
- According to the representation in Figure 1, 16 segmental longitudinal strains are shown and these are grouped into six sections. In this sense, and I consider it to be the most important point, the records represent 16 pseudoreplicas per patient. Therefore, the correlation analysis between SpLs and SLGE shows the associated dispersion but due to the pseudoreplicates a clear pattern associated with the segments is not observed and therefore the Pearson correlation statistical test applied in this way is not functional and neither inferential. There are two ways to analyze this data:
- Obtain the average values ​​and standard errors for each segment of the 30 patients and graph them in x and y with the associated standard errors for SpLs and SLGE (see paper). Therefore, 16 linear/or non-linear regressions can be estimated to be able to make inferences.
Rensink, R. A. (2017). The nature of correlation perception in scatterplots. Psychonomic Bulletin & Review, 24, 776-797.
- Perform a nested ANOVA analysis, where the average response can be compared at different levels between the 16 segments and six sections. This analysis resolves the presence of pseudoreplicates, which is a recurring situation for obvious reasons in medical or biomedical studies, although it is not exclusive to the area. To help in this process, the records of the 16 segments must be nested in the six sections and both as simple nesting or double nesting in the patients. Here the reason for the comparative and associated hypothesis games of the ANOVA lies in highlighting the importance and contrast between segments and sections. It is important to indicate that the records according to the statistical model may require that the nesting be random. Annex pdf of article and citation of the book to be reviewed and verify the assumptions of normal error distribution and homogeneity of variances.
Lazic, S. E. (2010). The problem of pseudoreplication in neuroscientific studies: is it affecting your analysis? BMC Neuroscience, 11, 1-17.
Zar, J. H. (2019). Biostatistical analysis. Pearson Education, Incorporated.
Applying this type of analysis is a way of innovating from the methodological perspective, in addition to not incurring errors by applying tests incorrectly and violating the assumptions. In this sense, it also allows the reproducibility of the study and technical repeatability.
We appreciate your suggestions regarding the handling of pseudoreplicates in our data. While we acknowledge that the presence of pseudoreplicates may influence the interpretation of Pearson correlation analysis, we believe that the scope of current study does not extend to the alternative approaches you’ve proposed. We would be happy to consider incorporating these approaches in future studies as we continue to refine our methodology.
- Check that the references are located in the format requested by the Journal.
The format has been corrected.
Reviewer 4 Report
Comments and Suggestions for Authors
The idea of ​​establishing a correlation between cardiac magnetic resonance late gadolinium enhancement (CMR-LGE) and speckle tracking echocardiography (STE) is particularly interesting, being able to indicate which of the two medical explorations is more useful in evaluating the patient with myocarditis.
Even though the number of patients enrolled in the study is quite small, the research can represent a starting point for the future research.
The methodology is well described, meeting the rigors of a valuable study, the results are well synthesized, and the discussions manage to emphasize the importance of the current research.
The references are up-to-date.
Author Response
We thank the reviewers for their positive remarks. The manuscript has been duly reviewed and corrected based on the constructive suggestions by all reviewers, thereby we believe that the final version would represent scientific message more lucidly.
Please find the detailed point by point responses below with all the corrections highlighted in Yellow in the re-submitted files.
Reviewer # 4:
- The idea of ​​establishing a correlation between cardiac magnetic resonance late gadolinium enhancement (CMR-LGE) and speckle tracking echocardiography (STE) is particularly interesting, being able to indicate which of the two medical explorations is more useful in evaluating the patient with myocarditis.
Thank you for your positive feedback. Indeed, understanding the relationship between these two modalities is of great clinical importance, particularly in patients with Acute myocarditis.
- Even though the number of patients enrolled in the study is quite small, the research can represent a starting point for the future research.
We recognize that the sample size is not large due to data collection from a single center, but we believe that it was sufficient to draw comparable conclusions and can serve as a starting point for future research.
- The methodology is well described, meeting the rigors of a valuable study, the results are well synthesized, and the discussions manage to emphasize the importance of the current research.
We are pleased that the rigorous approach we took in designing and conducting this research has been recognized.
- The references are up-to-date.
Thank you for your positive observation regarding the up – to – date references used in this article.
Round 2
Reviewer 2 Report
Comments and Suggestions for Authors
The article may be published as it is
Author Response
Dear Reviewer,
We would like to share our sincere gratitude towards the constructive remarks that helped us improve the scientific context of our study.
Kind regards
Reviewer 3 Report
Comments and Suggestions for Authors
Dear authors,
Thank you very much for addressing the comments. However, I will only insist on pseudoreplication. The lack of independence does affect the correlation coefficients and again I will indicate that a robust option that avoids this problem is a nested ANOVA. However, you can solve the graph in Figures 3 and 4 by applying a correlation to the average values ​​for intervals of SpLS and SLGE (see Word file).
I do not consider that the justification you indicate has support and in that sense, you should justify it in the statistical analysis section. See references
"Pseudoreplication occurs when observations are not statistically independent, but treated as if they are. This can occur when there are multiple observations on the same subjects, when samples are nested or hierarchically organised, or when measurements are correlated in time or space"
Di Credico, G., Pelucchi, S., Pauli, F., Stringhi, R., Marcello, E., & Edefonti, V. (2025). Nonlinear mixed-effects models to analyze actin dynamics in dendritic spines. Scientific Reports, 15(1), 5790.
Eisner, D. A. (2021). Pseudoreplication in physiology: More means less. Journal of General Physiology, 153(2), e202012826.
Lazic, S. (2009). Pseudoreplication invalidates the results of many neuroscientific studies.
Lazic, S. E. (2022). Genuine replication and pseudoreplication. Nature Reviews Methods Primers, 2(1), 23.
Lazic, S. E., Mellor, J. R., Ashby, M. C., & Munafo, M. R. (2020). A Bayesian predictive approach for dealing with pseudoreplication. Scientific Reports, 10(1), 2366.
On the other hand, I understand that you verified the assumptions and must integrate the values ​​of the K-S test and its corresponding P value, and you also mention that the "majority" comply, but it is not a matter of "majority" because all the variables have to comply with the assumption.
I am concerned about your sentence, I quote:
"The hypotheses associated with the Pearson correlation were tested based on assumptions of normality, and statistical significance was defined as a p-value ≤ 0.05"
This is incorrect since the value of P must be equal to or greater than 0.05. I do not consider it to be just an error of misplacing the symbol, because here lies the meaning of accepting or rejecting the set of hypotheses of the test.
Therefore, it is necessary to once again carry out an analysis of your justifications supported by evidence and you have an option to resolve your correlation analysis.
Best regards,

Make fine adjustments to the language style
Author Response
Dear Reviewer,
The response to your queries have been added in the document attached. We hope these exhaustive corrections shall satisfy your queries and improve the scientific context of the research. We thank you again for your constructive remarks.
Best regards.

Round 3
Reviewer 3 Report
Comments and Suggestions for Authors
Dear Authors,
You are very kind in indicating and making the adjustments to the manuscript. There are only a few points left to clarify, include, and justify.
1. In lines 161-3, you indicate that you performed the adjustment of the nested ANOVA model, so you should include the model with which you analyzed, an aspect that I had previously indicated.
2. If you performed the ANOVA with the mixed adjustment, the values ​​of the F ratio and its respective P value are not indicated in any paragraph of the results. Please include and describe them in the results, since the effect of a mixed variable provides an F value, and obviously the contrast of the factors that you tested, which are the estimated values ​​of the ANOVA table. Please incorporate results that I did not place in the results.
3. Figure 3 shows the correlation between the residuals and they found a P value>0.05, this means that there is no correlation. Be careful, because the model adjustment of ro ​​shows a negative association, and they are contradicting each other. Also, it is important because in lines 211 to 15, they show the values ​​of ro. So a value of P<0.5 indicates association, review and correct.
I hope you can adjust your manuscript and the results,
Best regards,
Author Response
Respected Reviewer,
Please find the response attached in the Word document.
